# Dietary Polysaccharide-Rich Extract from Noni (*Morinda citrifolia* L.) Fruit Modified Ruminal Fermentation, Ruminal Bacterial Community and Nutrient Digestion in Cashmere Goats

**DOI:** 10.3390/ani13020221

**Published:** 2023-01-06

**Authors:** Qingyue Zhang, Yanli Zhao, Yinhao Li, Xiaoyu Guo, Yongmei Guo, Guoqiang Ma, Xiaoshuai Liang, Sumei Yan

**Affiliations:** Key Laboratory of Animal Nutrition and Feed Science at Universities of Inner Mongolia Autonomous Region, College of Animal Science, Inner Mongolia Agricultural University, Hohhot 010018, China

**Keywords:** noni, polysaccharide, ruminal fermentation, bacterial community, digestibility, cashmere goat

## Abstract

**Simple Summary:**

Due to the seasonal imbalance and limited nutrients in pastures available for grazing in China, cashmere goats are often raised in a confined yard-feeding system, which may lead to metabolic disease from a lack of green pastures. The rumen is an important organ for nutrient metabolism in ruminants. Noni (*Morinda citrifolia* L.) could promote in vitro ruminal fermentation in lactating dairy goats. In addition, Noni fruit polysaccharides could alleviate inflammatory bowel disease by regulating the intestinal microbial composition in mice. Therefore, noni fruit polysaccharides may help ruminants to metabolize nutrients. The aim of this experiment was to characterize the effects of polysaccharide-rich noni fruit extract (NFP) on ruminal fermentation, ruminal microbiota and nutrient digestion in cashmere goats combining in vitro and in vivo techniques. The results showed that NFP depressed protozoa, improved N utilization and enhanced ruminal fermentation in vitro, which was better when the dosage was 0.40%. This dosage yielded similar results in vivo and promoted nutrient digestibility. In addition, the high proportion of Firmicutes/Bacteroides might be compensation for the decrease in protozoa, and the increase in volatile fatty acid concentration might be associated with the greater abundance of Ruminococcus_1. The Rikenellaceae_RC9_gut_group might have a negative effect on ruminal N utilization.

**Abstract:**

In two consecutive studies, we evaluated the effects of polysaccharide-rich noni (*Morinda citrifolia* L.) fruit extract (NFP) on ruminal fermentation, ruminal microbes and nutrient digestion in cashmere goats. In Exp. 1, the effects of a diet containing NFP of 0, 0.1%, 0.2%, 0.4% and 0.55% on in vitro ruminal fermentation at 3, 6, 9, 12 and 24 h were determined, whereas in Exp. 2, fourteen cashmere goats (46.65 ± 3.36 kg of BW ± SD) were randomly assigned to two treatments: the basal diet with or without (CON) supplementation of NFP at 4 g per kg DM (0.4%). The in vitro results showed that NFP linearly increased concentrations of volatile fatty acids (VFA), quadratically decreased ammonia-N concentration, and changed pH, protozoa number, gas production and the microbial protein (MCP) concentration, and was more effective at 0.4% addition, which yielded similar results in ruminal fermentation in Exp. 2. In addition, NFP increased the apparent digestibility of dry matter and crude protein and the abundance of Firmicutes, and reduced the abundance of Bacteroides and Actinobacteria. Ruminococcus_1 was positively associated with VFA concentration. The Rikenellaceae_RC9_gut_group was positively correlated with protozoa and negatively correlated with MCP concentration. Thus, NFP has potential as a ruminal fermentation enhancer for cashmere goats.

## 1. Introduction

The Inner Mongolia cashmere goat is one of the most famous breeds with dual-purpose use for cashmere and meat production in China. The cashmere fiber and meat it produces are both of high economic value [1]. This breed is distributed in the western part of Inner Mongolia, and the main production area (latitude 37°35′24″~40°51′40″ N, longitude 106°42′40″~111°27′20″ E) belongs to a typical temperate continental climate. Therefore, cashmere goats are used to grazing on semi-desert and desert grassland with supplementary feeding in the cold winter and early spring seasons [2]. However, due to the seasonal imbalance of nutrients in pastures and the limited deteriorated pasture supply, semi-intensive and intensive yard-farming systems for cashmere goats have been gradually adopted. Goats are fed in fenced yards without grazing under intensive feeding systems. However, some researchers have concerns regarding intensive feeding systems, including poor animal health and metabolic diseases, such as a higher prevalence of caseous lymphadenitis in intensively farmed sheep than in non-intensively farmed sheep [3]. The rumen is one of the most important organs for nutrient digestion and metabolism in ruminants. The efficiency of nutrient utilization by ruminants depends to a large extent on the balance of fermentation products in the rumen, which is ultimately controlled by the types of ruminal microorganisms [4]. Thus, regulation of ruminal fermentation in cashmere goats under intensive rearing conditions is important for the health of ruminants.

Noni (*Morinda citrifolia* L.) is a tropical plant with a long history of both medicinal and edible purposes in Polynesia. There are also abundant noni germplasm resources in south China. In recent decades, noni fruit has attracted lots of research attention because of its extensive pharmacological and biological functions including anti-inflammatory [5], antioxidant [6], anticancer [7] and immune modulatory effects [8]. Noni waste from noni juice was able to support microbial protein synthesis and fermentability in lactating dairy goats in vitro [9]. It could also improve the C18:1 fatty acid concentration in the milk of Holstein dairy cows [10]. Moreover, dietary supplementation of noni fruit has been found to promote feed conversion efficiency in beef cattle [11] and broiler chickens [12].

Fresh, ripe noni fruit is not easy to store and transport. Therefore, research on noni fruit processing, such as juicing and extraction, has attracted the attention of many researchers [8,13,14,15]. Among the effective compounds of noni, polysaccharides have better hydrophilicity and a variety of biological activities, such as enhancing the cell-mediated immune response [16], anti-inflammation [17] and antitumor [18]. Noni fruit polysaccharides could alleviate inflammatory bowel disease by attenuating disruption of the microbial composition and upregulating the content of acetic acid, propionic acid and butyric acid in mice [17]. Research on the regulation of ruminal fermentation or the ruminal bacterial community in ruminants by noni fruit polysaccharides is lacking so far. However, dietary supplementation of polysaccharides from other plants, such as Astragalus [19,20] and Momordica charantia [21], could effectively regulate ruminal fermentation in vivo or in vitro; in addition, the polysaccharides from Astragalus [22] and Glycyrrhiza uralensis Fisch [23] could increase body weight gain in calves [22] or in mice [23], respectively.

Based on the functions of antioxidants and the regulation of the immune and gut microbiota of noni fruit in mice and other species of animals, as well as the modulation of ruminal fermentation and growth performance by other plant polysaccharides, we hypothesized that noni fruit polysaccharides could regulate ruminal fermentation and ruminal bacterial communities, and consequently affect the metabolism of nutrients and their growth performance in cashmere goats. Therefore, this study is designed to explore the effects of adding polysaccharide-rich extract from noni fruit (NFP) to cashmere goat diets on ruminal fermentation, nutrient digestion and ruminal bacterial community. Therefore, the aim was to elucidate the relationship between ruminal microbiota and the ruminal fermentation of its host, and to discuss the ruminal microbial adaptation to the diet containing NFP, providing a basis for the utilization of noni fruit resources in the field of feeds. It also provides new ideas for the research of the microbial profile in cashmere goat breeding.

## 2. Materials and Methods

The study consisted of two in vitro ruminal fermentation experiments (Exp. 1) and an animal rearing experiment (Exp. 2). The goat experiment was carried out in the Experimental Farm of Inner Mongolia Agricultural University (Hohhot, China). The use of the animals were approved by Animal Ethics and Welfare Committee of Inner Mongolia Agricultural University in accordance with the Laboratory Animal Sciences and Technical Committee of the Standardization Administration of China, and performed under the national standard Guidelines for Ethical Review of Animal Welfare [24].

### 2.1. Preparation of Polysaccharide-Rich Fraction of Noni Fruit

Ripe noni fruit were provided by noni fruit planting base in Wuzhishan City, Hainan Province, China. After oven drying at 65 °C, the noni slices were ground into powder with the aid of an electric pulverizer (CH-200A, Chenhe Shengfeng Industry and Trade Co., Ltd., Yongkang, China) to pass a 1 mm screen. The noni powdered fruit were extracted with distilled water (1:25 ratio of raw material to water, *w/v*) by soaking for 12 h at 70 °C. Then, the filtrate after filtration (using filter paper with a maximum pore size of 15~20 μm) was concentrated to one-tenth of the volume under reduced pressure in a rotary evaporator at 65 °C. The concentrated solution was mixed with absolute ethanol (1:4, *v/v*) and then left to stand at 4 °C for 48 h to precipitate the polysaccharides. The polysaccharide-rich extract of noni fruit (NFP) was obtained by centrifugation (1200× *g*, 15 min) and lyophilized using a freeze dryer (Biosafer-10 C, Safer (China) Co., Ltd., Nanjing, China). The yield of NFP was 11.02%. We used ultraperformance liquid chromatography (UPLC; Shim-pack UFLC Shimadzu CBM30A)–electrospray ionization tandem mass spectrometry (MS, Applied Biosystems 6500 Q TRAP, UPLC-ESI-MS/MS) system to identify the structures of the active ingredients of NFP, according to the method by Li et al. [25]. The results are shown in Table 1.

### 2.2. Experiment 1: In Vitro Batch Fermentation

#### 2.2.1. Experimental Design and Diets

Two in vitro experiments were performed in (1) 120 mL serum bottles to determine ruminal fermentation variables, (2) 250 mL incubation bottles (for ANKOM RFS system, ANKOM Technology, New York, NY, USA) to determine gas production (GP). Two experiments were carried out as a completely randomized design with the same treatment diets: control diet (70:30 forage to concentrate) without supplement (CON), control diet with NFP at 0.10%, 0.20%, 0.40% or 0.55% (P0.1, P0.2, P0.4 and P0.55, respectively) based on the whole diet (on percent of dry matter basis). The feed ingredients and chemical composition of the control diet are shown in Table 2 and Appendix A.

#### 2.2.2. Inoculum and Substrates

Ruminal fluid was collected from 6 castrated male, 2.5-year-old goats (41.02 ± 2.30 kg of BW ± SD, with the strain of Inner Mongolia Albas white cashmere goat). Goat donors were fed with total mixed ration (TMR), which had the same nutrient composition as the control diet in Table 2, consisting of 30% concentrate and 70% roughage, twice daily at 800 and 1500 h. They were given free access to drinking fresh water. The buffer solution was prepared according to Menke et al. [26], prewarmed at 39 °C and purged with carbon dioxide (CO_2_). Approximately 0.3 L of ruminal digesta was collected from each goat before morning feeding using an esophageal tube passed through the mouth. Ruminal fluid was strained through 2 layers of cheesecloth and then mixed with buffer solution at a ratio of 1:2 (vol/vol) purged by CO_2_. Before in vitro experiments, feed ingredients used to make up the substrate were ground through a pulverizer (CH-200A, Chenhe Shengfeng Industry and Trade Co., Ltd., Yongkang, China) with a mesh size of 1 mm, and then mixed as described in Table 2 and stored at −20 °C until incubations were conducted.

#### 2.2.3. In Vitro Incubation and Sampling

Exactly 1.0000 g (±0.0002 g) of substrates were fed into 120 mL serum bottles or 250 mL incubation bottles, and then appropriate addition of NFP was weighed. A 60 mL mixture of ruminal fluid from goat donors and medium solution were added into each bottle with CO_2_ purge. After all additions, the bottles were flushed with CO_2_ and closed immediately with stoppers (for serum bottles) or modules (for incubation bottles), shaken and incubated in a shaking incubator (MAXQ 4000, ThermoFisher Scientific (China) Co., Ltd., Shanghai, China) at 39 °C for 3, 6, 9, 12 or 24 h, respectively (serum bottles), and 24 h consecutively (incubation bottles). The experiments, using both serum bottles and incubation bottles, were repeated two times on separate days with three replicates run at once.

The fermentation reaction was stopped by placing bottles in cold ice. For serum bottles, pH value of the fermentation solution was immediately measured using a portable pH meter (CT-6023; Kedida Electronics Co., Ltd. Shenzhen, China); and then the fermented solution was filtered through 2 layers of cheesecloth. Three aliquots of 1 mL samples were mixed with 4 mL of 3.5% formalin containing 8.0 g/L sodium chloride and 0.6 g/L methyl green stored at 4 °C for microscopic counting of protozoa; two aliquots of 0.5 mL filtrate were mixed with 4.5 mL of 0.2 mol/L hydrochloric acid to fix nitrogen for the determination of ammonia-N (NH_3_-N); two aliquots of 4 mL filtrate were mixed with 1 mL of 25% metaphosphoric acid solution for the determination of volatile fatty acids (VFA); the residual filtered liquid was collected for microbial protein (MCP) analysis. Samples for VFA, NH_3_-N and MCP were stored at −20 °C until analysis. For incubation bottles, volume of GP was measured (ANKOM RFS system, ANKOM Technology, New York, NY, USA) at 0, 3, 6, 9, 12 and 24 h of incubation.

#### 2.2.4. Calculation

Gas accumulation pressure at the top of incubation bottles was measured with a pressure transducer connected to a digital reader. Psi unit conversion was performed according to manufacturer’s instructions:V_x_ = V_j_ × P_psi_ × 0.068004084,(1)
where V_x_ is the cumulative gas production (mL); V_j_ is the volume of the space above liquid surface in the incubation bottle (i.e., 250 − 60 = 190, mL); P_psi_ is the gas accumulation pressure recorded by the digital reader (lbf/square inch); 0.068004084 is the constant term when converting psi to standard atmospheric pressure.

### 2.3. Experiment 2: Rearing Experiment

#### 2.3.1. Animals, Diets and Experimental Setup

Fourteen healthy, castrated male, 2.5-year-old goats (46.65 ± 3.36 kg of BW ± SD, with the same strain as ruminal fluid donor goats in Exp. 1) were used as experimental animals in a randomized complete block design. The goats were weighed and assigned into seven blocks. Each block had two goats with similar weight, and those two goats within each block were randomly assigned to one of the two experimental treatments: basal diet without supplementation (CON) or with supplementation with 4 g/kg DM (0.40% NFP) (according to the results of Exp. 1). Each treatment had 7 goats. The goats had individual water and feeder access and were placed into an individual pen (width = 100 cm, length = 150 cm and height = 100 cm) with elevated wooden floor (30 cm from the ground) with gaps to drain the urine and feces. The experiment lasted 22 d, including 2 wk for adaptation and 8 d for sampling and measurement. The feed ingredients and chemical composition of the basal diet were same as the control diet in Exp. 1 (Table 2). All ingredients were mixed in TMR, and 4 g/kg DM NFP was mixed with 200 g of TMR, then top-dressed on the rest of TMR in the morning. Respective treatment diets were offered to the animals twice daily at 800 and 1500 h.

#### 2.3.2. Sampling and Measurements

The goats were weighed continuously before morning feeding on the first two days at the beginning and the last two days at the end of the experiment and the average daily weight gain was calculated (ADG = (final weight − initial weight)/feeding days). The amounts of feed supplied and the refusals (about 5%~10%) were weighed and recorded for the quantification of intake, and samples of feed and refusals were dried in an oven at 65 °C for 48 h and ground in a pulverizer (CH-200A, Chenhe Shengfeng Industry and Trade Co., Ltd., Yongkang, China) to pass a 1 mm screen. During the 8-day experimental measurement period, all goats were fitted with a fecal collection bag and the output of feces was recorded daily. Feces were sampled by 1/5 of wet weight and stored at −20 °C. Samples of 8 single days during the measurement period were later thawed and mixed, and a subsample was dried at 65 °C for 72 h and ground to pass a 1 mm screen.

For the experimental period, ruminal fluid sampling was collected at 1 h before morning feeding (700 h) and 6 h later (1400 h) on the last two days (21 and 22 d), respectively. Ruminal contents (primarily the liquid phase) were collected using a sampler that was inserted via the mouth into the rumen. The first 100 mL of each sample were discarded to prevent saliva and cross contaminations, and stomach tubes were washed with warm water between collections [27]. The pH was immediately measured using a portable pH meter (CT-6023; Kedida Electronics Co., Ltd. Shenzhen, China). Then approximately 100 mL ruminal fluid was strained through four layers of cheesecloth. Three aliquots of 2 mL filtrate only collected at 1 h before morning feeding on 21 and 22 d were placed into cryogenic vials (Corning, New York, NY, USA), shock frozen in liquid nitrogen and stored at −80 °C until microbial DNA extraction. Samples collected from 700 and 1400 h on 21 and 22 d used for analyses of NH_3_-N, VFA and MCP concentrations and microscopic counting of protozoa, using the same procedure as in Exp. 1 to retain.

### 2.4. Determination of Chemical Composition and Calculation of Apparent Nutrient Digestibility

Samples of feed and feces were analyzed for DM (method 930.15), ether extract (EE) (method 973.18), CP (method 976.05), calcium and phosphorus (method 935.13) according to the methods by the Association of Official Analytical Chemists [28]. Neutral detergent fiber (NDF) and acid detergent fiber (ADF) were determined according to methods of Van Soest et al. [29] with an Ankom 200I Fiber Analyser (Ankom Technology Co., New York, NY, USA), and sodium sulfite and heat stable alpha-amylase were used and expressed exclusive of residual ash.
Apparent nutrient digestibility = [diet nutrient content (%) × DMI − fecal nutrient content (%) × fecal output (kg)]/diet nutrient content (%) × DMI (kg)

### 2.5. Determination of Ruminal Fermentation Variables

Concentrations of ruminal VFA (acetate, propionate, butyrate, iso-butyrate, valerate, iso-valerate) were determined by gas chromatography (GC-2014ATFSPL, Shimadzu, Kyoto, Japan; film thickness of the capillary column, 60 m × 0.25 mm×0.50 μm; column temperature, 180 °C; injector temperature 220 °C; detector temperature, 250 °C) according to Erwin et al. [30] with some modifications. After thawing at 4 °C, ruminal fluids from two days were mixed and then centrifuged at 2500× *g* for 15 min at 4 °C and 0.2 mL of metaphosphoric acid solution (250 g/L) containing 2 g/L 2-ethyl butyrate was added to 1 mL supernatant. The mix was vortexed and centrifuged at 10,000× *g* for 20 min at 4 °C and 1 µL of supernatants were injected into the gas chromatography for analysis. The concentration of NH_3_-N was measured using the phenol hypochlorite colorimetric method as described [31]. The microbial protein was measured by Lowry’s method [32]. The methods of microscope counting of ruminal protozoa were according to Dehority [33].

### 2.6. DNA Extraction and Sequencing

Bacterial community genomic DNA was extracted from the mixed samples of ruminal fluid collected over two days using FastDNA^®®^ Spin Kit for Soil (MP Biomedicals, Santa Ana, CA, USA) according to manufacturer’s instructions. The DNA concentration and purity were determined with NanoDrop 2000 UV-vis spectrophotometer (Thermo Scientific, Wilmington, NC, USA), and DNA quality was evaluated by 1% agarose gel electrophoresis. The hypervariable region V3 to V4 of the bacterial 16S rRNA genes was amplified with the primer set 338 forward (5′-ACTCCTACGGGAGGCAGCAG-3′) and 806 reverse (5′-GGACTACHVGGGTWTCTAAT-3′) using an ABI GeneAmp^®®^ 9700 PCR thermocycler (ABI, Santa Ana, CA, USA). The PCR amplification of 16S rRNA gene was performed as follows: initial denaturation at 95 °C for 3 min, followed by 27 cycles of denaturing at 95 °C for 30 s, annealing at 55 °C for 30 s and extension at 72 °C for 45 s, and single extension at 72 °C for 10 min, and end at 4 °C. The PCR reactions were performed in triplicate 20 μL mixture containing 4 μL of 5× TransStart FastPfu buffer, 2 μL of 2.5 mM dNTPs, 0.8 μL of each primer (5 μmol/L), 0.4 μL of TransStart FastPfu DNA Polymerase, 10 ng of template DNA. The amplified products were extracted from 2% agarose gel, further purified using the AxyPrep DNA Gel Extraction Kit (Axygen Biosciences, Union City, CA, USA) and then quantified using Quantus™ Fluorometer (Promega, Madison, WI, USA) according to manufacturer’s instructions. Subsequently, purified amplicons were pooled in equimolar amounts and sequenced on an Illumina MiSeq platform (Illumina, San Diego, CA, USA) for paired-end reads of 300 bp at Majorbio Bio-Pharm Technology Co. Ltd. (Shanghai, China) according to standard protocols.

### 2.7. Sequence Processing and Data Analysis

The generated raw sequencing reads were quality-filtered by Trimmomatic and merged by FLASH with the following criteria: (i) the reads were truncated at any site receiving an average quality score of <20 over a 50 bp sliding window, and reads containing ambiguous characters were also discarded; (ii) sequences with overlaps longer than 10-bp were merged according to their overlap with mismatches ≤ 2 bp; (iii) primers matching allowed 2-nucleotide mismatching, and reads containing ambiguous bases were removed. Operational taxonomic units (OTUs) were clustered with a 97% similarity cut-off using UPARSE (version 7.1 http://drive5.com/uparse/ (accessed on 30 September 2013)), and chimeric sequences were identified and removed. The taxonomy of each OTU representative sequence was analyzed by RDP Classifier (http://rdp.cme.msu.edu/ (accessed on 30 September 2016)) against the Silva (SSU128) 16S rRNA database using confidence threshold of 70% [34].

Taxonomic identification and comparisons were performed at the OTU levels of phylum, family and genus. Alpha diversity indexes (i.e., observed species, Chao1, ACE, Shannon, Simpson and Good’s coverage index) were calculated using MOTHUR (versionv.1.30.1) [35]. The rarefaction and Shannon curves were generated using vegan package in R [36]. Beta diversity was estimated by computing the unweighted UniFrac distance and visualized using principal coordinate analysis (PCoA), and the results were plotted using GUniFrac and ape packages in R [37,38].

### 2.8. Statistical Analysis

The data of ruminal fermentation variables in Exp. 1 (including GP) and 2 were analyzed using PROC MIXED of SAS (version 8.1, SAS Institute Inc., Cary, NC, USA). The statistical model used for analysis was yijk = μ + Li + Ej + LEij + εijk, where yij is the dependent variable, μ is the overall mean, Li is the fixed effect of diet (i = 1 to 5 in Exp. 1 and i = 1 to 2 in Exp. 2), Ej was the fixed effect of incubation time (Exp. 1, j = 1 to 5) or sampling time (Exp. 2, j = 1 to 2); LEij was also considered as fixed effects of the interaction of diet and incubation time (Exp. 1) or sampling time (Exp. 2), and εijk was the residual error. In Exp. 1, batch effects were also added to statistical models.

Data of nutrient digestibility and growth performance in Exp. 2 were analyzed using *t*-test in SAS 8.1. Principal coordinate analysis (PCoA) was performed using the weighted UniFrac distance using R. Non-parametric multivariate analysis of variance (Adonis) was performed on the weighted UniFrac distances to assess the significance of differences in bacterial community structure between groups. Student’s *t*-test was used to analyze differences in diversity indexes as well as relative abundances at phylum, family and genus level. The results are presented as the mean and SEM. The number of observations for each mean value was seven (n = 7).

For the above statistical analyses, significance was declared at *p* < 0.05 and trends at 0.05 < *p* < 0.10. Spearman correlation was used to correlate ruminal fermentation variables with the 16 most relatively abundant bacterial genera using R (pheatmap package). Only correlations with |R| > 0.5 and *p* < 0.01 for the linear model were considered as being significant.

## 3. Results

### 3.1. Experiment 1: In Vitro Ruminal Fermentation

Incubation time had an effect on pH, NH_3_-N concentration, number of protozoa, MCP concentration and GP (*p* < 0.001) (Table 3). There was no significant difference among treatments at 0 h (data for 0 h are not shown in the table). With the increase in incubation time, pH gradually decreased, GP gradually increased and then increased rapidly from 9 to 12 h. With increasing incubation time, the MCP concentration and the number of protozoa first increased and then decreased, both reaching the peak at 12 h of incubation. The NH_3_-N concentration went down first, then it flattened out, and then it went up and then down again. TRT×Hour interaction was not significant for ruminal fermentation variables other than pH and NH_3_-N concentration. For pH, the highest value was for CON when incubated for 6 h and the lower values were for NFP-containing diets when incubated for 24 h (Appendix A). In NH_3_-N concentration, the highest value was for CON when incubated for 12 h and the lower values appeared at 6 h and 9 h of incubation (Appendix A). NFP-containing diets reduced (*p* < 0.05) pH, NH_3_-N concentration and number of protozoa within 24 h of incubation. NFP quadratically affected NH_3_-N concentration (*p* = 0.004), and linearly (*p* < 0.05) and quadratically (*p* < 0.05) affected pH, MCP concentration, GP and number of protozoa, with extreme values observed at P0.4 for NH_3_-N concentration, MCP concentration, GP and number of protozoa. In addition, the number of protozoa in the P0.2, P0.4 and P0.55 groups was significantly lower than that in the CON group (*p* = 0.015).

Incubation time had an effect on all variables related to VFA (*p* < 0.001), but TRT×Hour interaction had no significant effect on them (*p* > 0.05) (Table 4). With the increase in incubation time, all indicators increased gradually except for A/P. The value of A/P went down first, then it flattened out and then it went down again. NFP linearly increased all VFA-related indicators (*p* < 0.05) except for A/P (*p* > 0.05), and quadratically affected propionate concentration (*p* = 0.047). Compared with CON, the concentrations of acetate and iso-butyrate were higher with P0.4 and P0.55 (*p* = 0.049), the concentrations of butyrate and total VFA (TVFA) were higher with all doses of NFP other than P0.1 (*p* < 0.05), and the propionate concentration was higher with all doses of NFP (*p* < 0.05). Among them, the concentration of iso-butyrate with P0.55 was higher than that with P0.1 and P0.2 (*p* = 0.004). In addition, compared with other diets (CON, P0.1 and P0.2), P0.4 led to the highest iso-valerate concentration (*p* = 0.027).

### 3.2. Experiment 2: Ruminal Fermentation

The TRT × Hour interaction was not obvious for ruminal fermentation variables (Table 5). The ruminal concentration of NH_3_-N and the number of protozoa were lower (*p* < 0.05) in NFP goats than that in CON goats. Compared with CON goats, the concentration of MCP, acetate, propionate, iso-butyrate, iso-valerate and total VFA were greater (*p* < 0.05) in NFP, and butyrate and valerate tended to be greater (*p* < 0.10), while the sampling time had an impact (*p* < 0.05) on ruminal variables. Compared with 1 h before feeding, pH, concentrations of NH_3_-N, MCP, iso-butyrate and iso-valerate, as well as acetate/propionate were reduced (*p* < 0.05) at 6 h after feeding, while the number of protozoa and the concentrations of acetate, propionate, butyrate, valerate and TVFA were increased (*p* < 0.05).

### 3.3. Growth Performance and Nutrient Digestion

Growth performance and apparent nutrient digestibility are presented in Table 6. The digestibility of dry matter and crude protein in goats of NFP were higher (*p* < 0.05) than in goats of CON, and the digestibility of Ca tended to be higher (*p* = 0.061). The BW was similar during the trial between the two treatments, but DM intake (DMI) and average daily gain (ADG) were greater (*p* < 0.05) in NFP goats compared with CON goats, and DMI:ADG tended to be greater (*p* = 0.065).

### 3.4. Sequencing Coverage and Bacterial Diversity

A total of 611,095 high-quality 16S rRNA gene sequences were obtained from 14 samples. The diversity and abundance in each sample are presented in Figure 1 and Table 7. Based on the 97% sequence identity, 287,924 bacterial sequences were assigned to 1753 OTUs. In the ruminal digesta, there were 1555 and 1574 OTUs obtained from the goats in CON and NFP, respectively, of which 1376 were shared OTUs and 377 were unique to one of the two treatments. The rarefaction and Shannon index curves generated from the OTUs showed that the data covered most of the diversity and new phylotypes (Figure 2), which indicated that the sequencing depth had captured a high sampling coverage, and it was unlikely to achieve more ruminal bacterial community diversity, even if the sequencing depth was further increased. The α-diversity indices (Table 7) indicated that supplementation did not exert predominant effects (*p* > 0.05) on Clean reads, Sobs, Shannon, Simpson, Ace and Chao indices. Although Good’s coverage between the two groups tended to be different (*p* = 0.064), they were both greater than 98.90%, indicating that the sequencing depth could cover most species and the sequencing data could be used for subsequent analyses.

### 3.5. Principal Coordinate Analysis

The samples clustered according to dietary treatment by using the weighted UniFrac similarity metric. Adonis analysis indicated a difference (*p* = 0.002) between treatments in their ruminal bacterial communities (Figure 3).

### 3.6. Microbial Composition Analysis

The most abundant eight phyla are listed in Figure 4 and Table 8, representing 97.87% (CON) and 97.45% (NFP) of the total microbiome. At the phylum level, the microbiota of ruminal digesta were dominated by Firmicutes and Bacteroidetes in goats of the two groups, followed by far less abundant Saccharibacteria. The relative abundance of Firmicutes was greater (*p* = 0.0002) in NFP goats than that in CON, and the relative abundances of Bacteroidetes and Actinobacteria were lower (*p* < 0.05) in NFP goats. The most abundant 14 families are listed in Figure 5 and Table 8, representing 90.10% (CON) and 90.90% (NFP) of the total microbiome. At the family level, the dominant families within the Firmicutes phylum consisted of Ruminococcaceae, Lachnospiraceae, Christensenellaceae and Erysipelotrichaceae, while the main families within the Bacteroidetes phylum were Prevotellaceae, Rikenellaceae, Bacteroidales_BS11_gut_group and Bacteroidales_RF16_group. The relative abundance of Ruminococcaceae was greater in NFP goats than that in CON (*p* = 0.0003). The relative abundances of Bacteroidetes, Rikenellaceae, Lachnospiraceae and Family_XIII were greater than (*p* < 0.05) those in CON goats compared with NFP goats, and the relative abundance of Bacteroidales_BS11_gut_group tended to be higher (*p* = 0.0595). Moreover, ruminal microbial composition and differences between treatments for the most abundant 16 genera are presented in Figure 6 and Table 8, representing 70.52% (CON) and 76.98% (NFP) of the total microbiome. Within Ruminococcaceae, the abundances of Ruminococcus_1, Ruminococcaceae_UCG-002 and norank_f__Ruminococcaceae were greater (*p* < 0.05) in NFP goats than those in CON, whereas the abundance of Ruminococcaceae_NK4A214_group was greater (*p* = 0.043) in the CON goats than that in NFP. Within other families, the abundances of Rikenellaceae_RC9_gut_group and Prevotellaceae_UCG-003 were greater (*p* < 0.05) in CON goats than those in NFP, and the abundance of norank_f__Bacteroidales_BS11_gut_group in CON goats tended to be greater (*p* = 0.0595), whereas the abundance of norank_f__Erysipelotrichaceae tended to be greater (*p* = 0.0588) in the NFP goats than that in CON.

### 3.7. Spearman Correlation Analysis between Ruminal Bacteria Abundance and Ruminal Fermentation Variables

Spearman correlation analysis was performed for the first 16 bacterial genera and the ruminal fermentation variables (Figure 7). A total of seven genera were related to ruminal fermentation variables (*p* < 0.01). The concentration of NH_3_-N was negatively correlated with Ruminococcus_1 and Ruminococcaceae_UCG-002, two bacteria of the Ruminococcaceae family, but was positively correlated with Christensenellaceae_R-7_group, while the number of protozoa was only negatively correlated with Ruminococcus_1, but was positively correlated with norank_f__Bacteroidales_BS11_gut_group and Rikenellaceae_RC9_gut_group. Rikenellaceae_RC9_gut_group was also negatively correlated with MCP. Ruminal TVFA concentration was only negatively correlated with Prevotellaceae_UCG-003 in Bacteroidetes phylum. Acetate, propionate and butyrate had a positive correlation with Ruminococcus_1. Propionate had a negative correlation with Rikenellaceae_RC9_gut_group. Acetate, butyrate and valerate had a negative correlation with Ruminococcaceae_NK4A214_group, while butyrate and valerate also had a negative correlation with Christensenellaceae_R-7_group. No distinct correlation was detected between pH and the abundance of any bacterial genus, or for A/P, iso-butyrate and iso-valerate.

## 4. Discussion

### 4.1. In Vitro Ruminal Fermentation of NFP Diets

The concentration of ruminal MCP is an important indicator for the comprehensive evaluation of protein utilization efficiency and microbial population [39]. NH_3_-N is the primary nitrogen source for MCP production. Moreover, many studies have shown that the decrease in NH_3_-N concentration and the increase in MCP concentration within a certain range imply the improvement in nitrogen utilization in the rumen [40,41]. The presence of protozoa increased the turnover and consumption of nitrogen in the rumen, and a reduction in ruminal protozoa could increase the flow of microbial N to the intestine [42]. In Exp. 1, the addition of NFP in vitro reduced NH_3_-N concentration and the number of protozoa, and tended to increase MCP concentration. This suggests that adding NFP improved the efficiency of N utilization in the rumen. These results were not identical to previous fermentation studies on noni waste in vitro. Diets containing seedless noni waste increased the concentrations of NH_3_-N and MCP in in vitro ruminal fermentation but without significant changes in protozoa numbers [9], and diets supplemented with noni juice extract waste enhanced NH_3_-N levels without having a significant effect on MCP [43]. The differences in the results of each test might be caused by the large loss of functional components from noni pomace and juice extraction waste in the process of fermentation or juice production.

The synthesis of high-quality MCP also requires a suitable pH to maintain the steady state of ruminal fermentation, and a pH between 6.0 and 7.0 is conducive to the synthesis of MCP [44]. In the current research, the pH decreased after NFP addition in Exp. 1 but not in Exp. 2, which might be related to the fact that only two time points were detected in the rearing experiment. It was also not ruled out that the fermentation bottles in vitro were closed and the nutrients could not be absorbed by the ruminal epithelium and circulated backwards. The pH values in vitro or in vivo were all within the range favorable for the synthesis of MCP.

VFA is the main energy source for ruminants and an important carbon framework source for ruminal microbes, and its concentration and composition are important indicators that reflect ruminal digestion and metabolism [45]. Ruminal GP is one of the major indicators of feed digestibility [46]. The increase in VFA concentration and GP in NFP diets in Exp. 1 suggested that NFP promoted ruminal fermentation in vitro. Similar results were obtained by Evvyernie et al. [9] and Anjani et al. [43]. The regression analysis results of ruminal fermentation variables in Exp. 1 showed that with the increase in NFP supplementation, concentrations of TVFA and each VFA except for propionate increased in a linear dependence manner, NH_3_-N concentration decreased in a quadratic dependence manner, and pH, protozoa number, GP and the concentrations of MCP and propionate changed in both linear and quadratic manners. Moreover, the indicators often changed significantly at the supplementation levels from 0.20% to 0.55%, and always reached extreme values at the supplementation level of 0.40%. This suggests that diets supplemented with from 0.20% to 0.55% NFP could promote rumen fermentation. In this rearing trial, 0.40% was chosen as the addition level of NFP in Exp. 2. Since the in vitro experiment conditions still differed from the in vivo environment of the organism, subsequent experiments are needed to verify whether other doses will have similar or better effects in vivo.

### 4.2. Ruminal Fermentation Variables In Vivo

Ruminal fluid pH usually oscillates depending upon, among other factors, meals and feeding times [47]. The pH of ruminal fluid collected at −1 h was higher than that at 6 h, probably because the VFA concentration was lower at this time, which was due to active uptake and reduced fermentable OM in the rumen [48].

Regarding the antiprotozoal function of NFP in this study, the results of association analyses showed that Rikenellaceae_RC9_gut_group (the relative abundance of CON group was 16.0%) had a positive correlation with protozoa and a negative correlation with MCP. This suggested that Rikenellaceae_RC9_gut_group might have a negative effect on the improvement of ruminal N utilization efficiency. Saponins have also been shown to kill protozoa by damaging the protozoa cell membrane in numerous studies [40,49]. This is most likely due to binding protozoal cell proteins and enzymes, as well as creating complexes with sterols in protozoal cell membranes [50]. In addition, the content of saponin in mature noni fruit is about 236.0 mg/100 g [51], and saponins are important terpenoid derivatives. NFP contained 1.29% terpenoids and their derivatives in this test, suggesting that a small amount of saponins contained in NFP may also reduce protozoa.

### 4.3. Apparent Digestibility of Nutrients, VFA and Ruminal Bacterial Community

Protozoa live on bacteria but can also ferment cellulose [52]. Newbold et al. [53] showed that the elimination of ruminal protozoa significantly decreased NDF (−20%) and ADF digestibility (−16%). Interestingly, in the present experiment, the number of protozoa was decreased by NFP, but the digestibility of NDF and ADF was similar between the two groups. It indicated that the antiprotozoal effect of NFP at this dose might not affect the degradation of fiber in cashmere goats. Bacteroidetes and Firmicutes are two predominant bacteria commonly found in the rumen. It is known that members of Bacteroidetes are the main ruminal microorganisms for degrading non-structural carbohydrates and non-fibrous polysaccharides, while Firmicutes are the main ruminal microorganisms for degrading structural carbohydrates [54]. In the present test, the abundance of Firmicutes was increased and the abundance of Bacteroidetes was decreased by NFP, although the kit-extracted DNA without a mechanical lysis step may generate a bias towards Gram-negative Bacteroidetes due to the fact that they are probably more susceptible to lysis than Gram-positive Firmicutes [55]. However, since the same method was used across samples, the present work remains valid. A high proportion of Firmicutes/Bacteroidetes is also a response to high-fiber food resources, which helps animals obtain more energy from feed [56]. Therefore, we suspected that the reduction in protozoa led to a compensatory increase in the proportion of Firmicutes/Bacteroidetes (niche replacement) in the rumen, which helped the host use more cellulose for energy. This also explains the results that NFP reduced the protozoa number but did not change the digestibility of fibers in the current trial. Moreover, fungi play an important role in rumen fiber digestion [57]. The abundance of fungi was not measured in this experiment; also, stomach tube sampling does not target particle-associated bacteria. Hence the effect of NFP on fiber digestion in the rumen needs to be further explored.

In Exp. 2, the concentration of VFA in the NFP group generally increased. The data showed that NFP yielded similar and significant promoting effects on DMI (1.63 vs. 1.35 kg DM/d (=20.7% increase)) and TVFA (42.2 vs. 34.7 mmol/L (=21.4% increase)). It indicated that NFP increased ruminal TVFA concentrations by increasing feed intake. Furthermore, the correlation analyses showed that the genera positively correlated with various VFA including the most abundant genus Ruminococcus_1 (the relative abundance of the NFP group was 26.0%), which was also negatively correlated with NH_3_-N concentration and the number of protozoa. More excitingly, the relative abundance of Ruminococcus_1 genus in goats of the NFP group was 10-fold higher than that of the CON group (NFP vs. CON = 26.0% vs. 2.67%). Ruminococcus_1 belong to the family Ruminococcus in Firmicutes. This family is known as saccharolytic bacteria, degrading pectin and cellulose, and important in the ruminal fermentation of dietary fibers [58] and are related to the production of various VFA such as acetate [59] and butyrate [60]. It is suggested that NFP promoted the degradation of fiber and other polysaccharides by increasing the abundance of Ruminococcus_1 in Ruminococcaceae, thereby increasing the concentration of VFA in the rumen. Furthermore, detailed physiological studies on Ruminococcus_1 are still largely lacking. The NFP substrate may be an ideal candidate for use in the enrichment of Ruminococcus_1, which is likely to isolate members of this genus in pure culture. In the future, our group will conduct an in-depth study on this matter.

### 4.4. Growth Performance

In this experiment, the addition of NFP increased ADG and tended to increase DMI:ADG. At present, there is still a lack of relevant reports on the regulatory effects of NFP on the growth performance of ruminants. However, the studies on other polysaccharides are similar to this trial. Diet including astragalus root extract rich in polysaccharides increased ADG and decreased the DMI:ADG of early weaned yak calves [22]. Feeding mouse Glycyrrhiza uralensis Fisch polysaccharides increased the feed efficiency of mice, and it was proportional to the given dose and the number of feeding days [23]. Regarding the growth-promoting effect of NFP in this test, it was likely that the significant increase in VFA (21.4%) in the rumen provided more energy for goats to digest nutrients, thereby improving the performance of cashmere goats with the NFP diet. On the other side, the oversize increase in DMI (20.7%) in the experimental group, suggests that sugars contained in NFP may have had the effect of promoting appetite. Moreover, non-nutritive sweeteners can separate sweetness from energy intake, and this separation may lead to incomplete energy compensation and, ultimately, increased food intake by activating food reward pathways [61]. The specific components of NFP appetite-promoting remains to be further studied. In addition, some studies have shown that adding astragalus polysaccharides to the diet has no obvious effect on the ADFI, ADG and feed efficiency of lambs [19]. This was probably related to factors such as the extraction method and administration method of the polysaccharides, as well as the composition and molecular active groups directly involved in the regulation of ruminal fermentation, which will affect the utilization effect of the polysaccharides. In the future, our research team will conduct more in-depth research on the molecular active groups in NFP that regulate ruminal fermentation.

## 5. Conclusions

Using the approach coupling in vivo and in vitro techniques, this study found clear evidence that NFP depressed protozoa, improved N utilization efficiency and enhanced ruminal fermentation, and the effect was better when the supplementation of NFP was 0.20%~0.55% in vitro. Feeding cashmere goats the diet containing 0.40% NFP yielded similar results in in vitro experiments. In addition, NFP promoted intake, modified the ruminal bacterial community and, to some extent, increased the apparent digestibility of nutrients and weight gain. In the present study, the high proportion of Firmicutes/Bacteroides in goats with a 0.40% NFP diet might be compensation for the decrease in protozoa, while the increase in various VFA concentrations in NFP goats might be associated with the more abundant Ruminococcus_1. Rikenellaceae_RC9_gut_group might have a negative effect on the improvement of ruminal N utilization efficiency.

## Figures and Tables

**Figure 1 animals-13-00221-f001:**
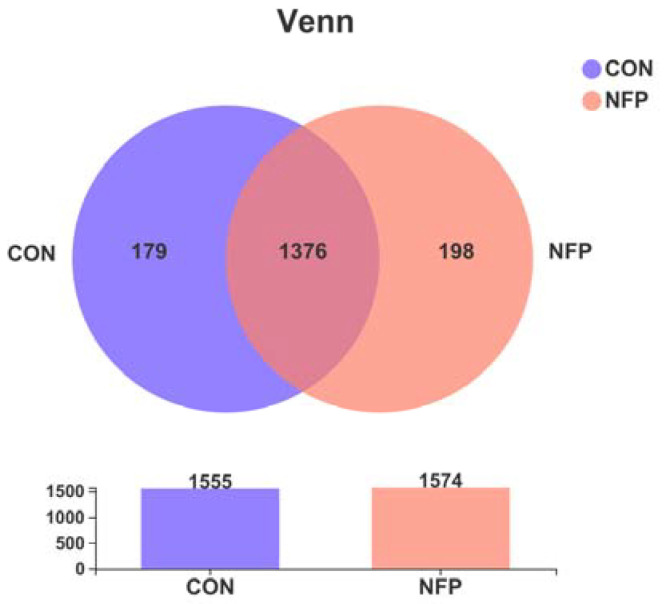
Common and distinctive bacterial OTU between the samples by Venn’s diagram in Exp. 2 (*n* = 7).

**Figure 2 animals-13-00221-f002:**
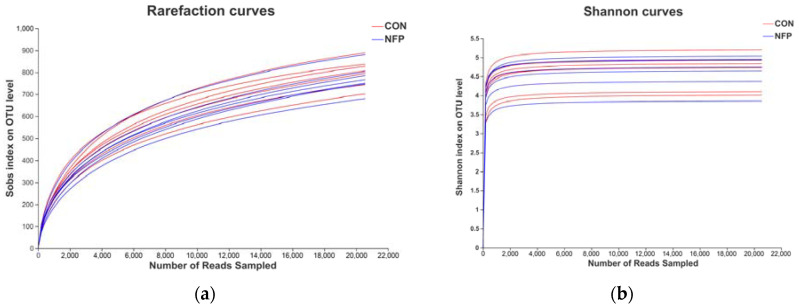
The rarefaction (**a**) and Shannon (**b**) index curves.

**Figure 3 animals-13-00221-f003:**
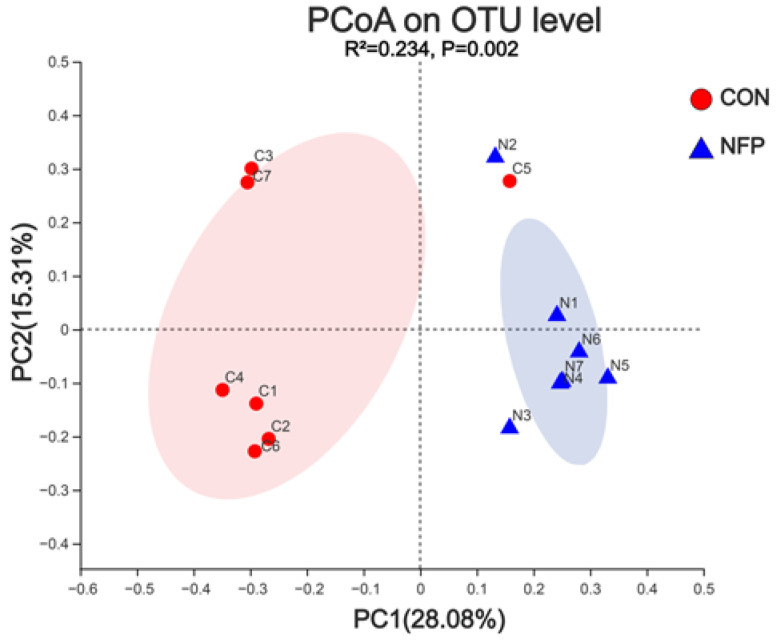
Principal coordinate analysis (PCoA) of ruminal microbiome in goats between CON and NFP group in Exp. 2 (*n* = 7). The pink part was the confidence interval of control (CON). The purple parts was the confidence interval of polysaccharide-rich extract of noni fruit (NFP).

**Figure 4 animals-13-00221-f004:**
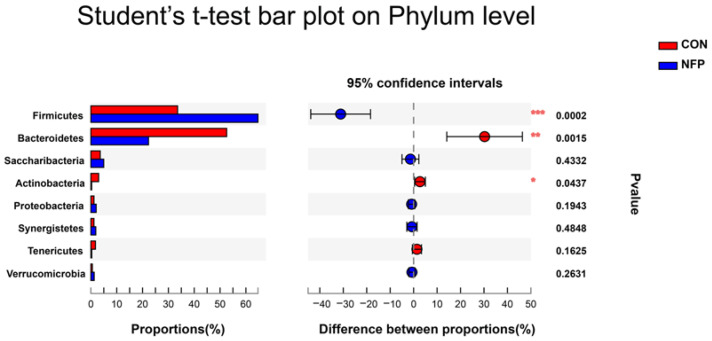
Statistical analysis of different bacterial taxa at the phylum level by Student’s *t*-test in Exp. 2 (* 0.01 < *p* ≤ 0.05, ** 0.001 < *p* ≤ 0.01, *** *p* ≤ 0.001).

**Figure 5 animals-13-00221-f005:**
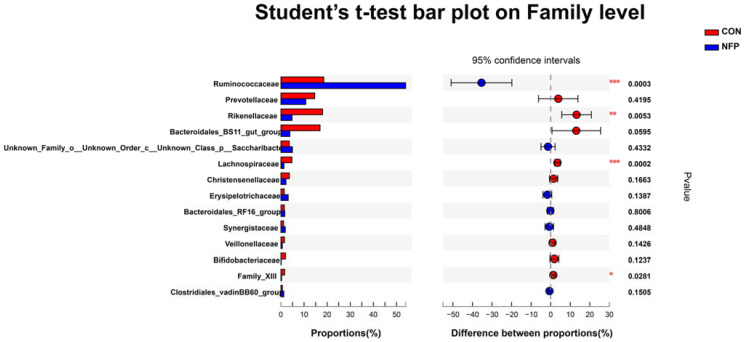
Statistical analysis of different bacterial taxa at the family level by Student’s *t*-test in Exp. 2 (* 0.01 < *p* ≤ 0.05, ** 0.001 < *p* ≤ 0.01, *** *p* ≤ 0.001).

**Figure 6 animals-13-00221-f006:**
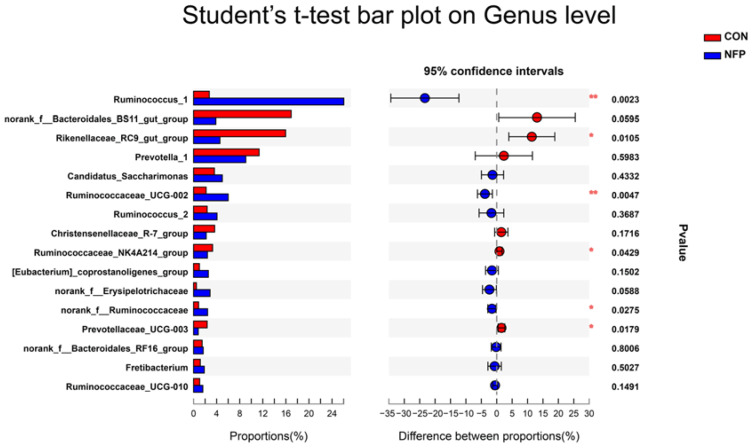
Statistical analysis of different bacterial taxa at the genus level by Student’s *t*-test in Exp. 2 (* 0.01 < *p* ≤ 0.05, ** 0.001 < *p* ≤ 0.01).

**Figure 7 animals-13-00221-f007:**
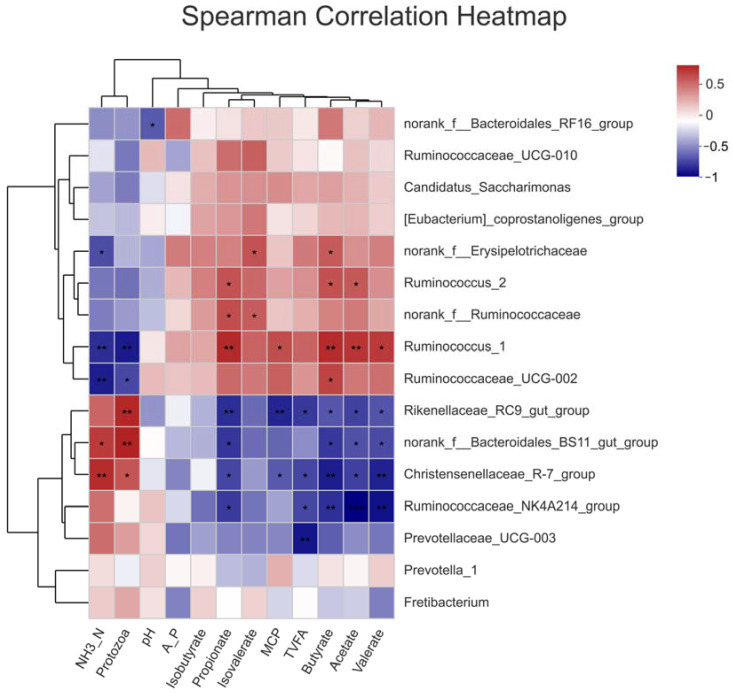
Spearman correlation analyses of the most abundant 25 bacterial genera with ruminal fermentation relevant variables in Exp. 2. Only genera with *p* ≤ 0.01 were concerned (* 0.01 < *p* ≤ 0.05, ** 0.001 < *p* ≤ 0.01, *** *p* ≤ 0.001).

**Table 1 animals-13-00221-t001:** Compound contents of NFP (%, DM basis).

Compounds	Content, %
Saccharides	45.47
Organic acids and their derivatives	21.37
Lipids	6.09
Alkaloids and their derivatives	6.14
Amino acids and their derivatives	7.06
Coumarins	1.36
Terpenes	1.18
Cholines	1.22
Nucleotides and their derivatives	1.21
Alcohols	1.13
Phenols and their derivatives	1.08
Vitamins and their derivatives	0.47
Salts	0.97
Others	5.27

**Table 2 animals-13-00221-t002:** Ingredients and nutrient composition of control diet.

Item	Content
Ingredient, g/kg air dry basis	
Millet straw	589.6
Alfalfa hay	29.6
Oat grass hay	80.8
Corn grain	145.8
Soybean meal	53.0
Distillers dried grains with solubles	33.0
Flax cake	53.0
Limestone meal	1.2
Calcium bicarbonate	1.2
Premix ^1^	5.0
Sodium chloride	3.0
Sodium bicarbonate	4.8
Nutrient composition	
Digestible energy, MJ/kg dry matter basis (DM) ^2^	9.95
DM, %, air dry basis	92.36
CP, %, DM	10.04
aNDFom ^3^, %, DM	54.71
ADFom ^4^, %, DM	30.98
EE, %, DM	2.24
Calcium, %, DM	0.63
Phosphorus, %, DM	0.29

^1^ Provided per kilogram of premix: iron 4 g, copper 0.8 g, zinc 5 g, manganese 3 g, iodine 30 mg, selenium 30 mg, cobalt 25 mg, vitamin A 600,000 IU, vitamin D 250,000 IU, vitamin E 1250 IU, vitamin K 180 mg, vitamin B1 35 mg, vitamin B2 850 mg, vitamin B6 90 mg, nicotinic acid 2200 mg, D-pantothenic acid 1700 mg, vitamin B12 3 mg, biotin 14 m, folic acid 150 mg. ^2^ Digestible energy was calculated based on the ingredients of the diet and their digestible energy content, not based on the actual dry matter intake. ^3^ Neutral detergent fiber assayed with a heat stable amylase and expressed exclusive of residual ash. ^4^ Acid detergent fiber expressed exclusive of residual ash.

**Table 3 animals-13-00221-t003:** The effect of NFP on in vitro ruminal pH, NH_3_-N, protozoa, MCP and GP within 24 h incubation.

Item	pH	NH_3_-N, mg/dL	Protozoa, 10^4^/mL	MCP, mg/dL	GP, mL
Treatment					
CON	6.67 ^a^	10.6 ^a^	6.11 ^a^	20.4	56.1 ^ab^
P0.1	6.52 ^b^	10.1 ^b^	5.48 ^ab^	23.3	60.3 ^ab^
P0.2	6.52 ^b^	9.97 ^b^	5.27 ^b^	24.7	60.7 ^a^
P0.4	6.53 ^b^	9.91 ^b^	5.03 ^b^	25.3	61.9 ^a^
P0.55	6.53 ^b^	10.1 ^b^	5.25 ^b^	24.5	60.5 ^a^
Incubation time, h					
3	6.78 ^A^	9.78 ^C^	0.37 ^E^	12.5 ^D^	21.6 ^E^
6	6.80 ^A^	6.99 ^D^	1.55 ^D^	15.2 ^CD^	43.6 ^D^
9	6.56 ^B^	6.96 ^D^	7.89 ^B^	17.2 ^C^	64.6 ^C^
12	6.53 ^B^	15.5 ^A^	12.6 ^A^	40.9 ^A^	76.8 ^B^
24	6.10 ^C^	11.6 ^B^	4.77 ^C^	32.4 ^B^	93.0 ^A^
SEM	0.017	0.155	0.231	1.25	1.51
*p*-value					
TRT	<0.001	0.011	0.015	0.085	0.112
linear	<0.001	0.051	0.007	0.012	0.031
quadratic	<0.001	0.004	0.027	0.039	0.048
Hour	<0.001	<0.001	<0.001	<0.001	<0.001
TRT × Hour	0.015	0.029	0.700	0.314	0.112

^a,b^ Without a common lowercase superscript letter indicates significant (*p* < 0.05) changes between treatments. ^A–E^ Without a common uppercase superscript letter indicates significant (*p* < 0.05) changes between hours.

**Table 4 animals-13-00221-t004:** The effect of NFP on in vitro ruminal volatile fatty acids within 24 h incubation.

Item	Acetate, mmol/L	Propionate, mmol/L	Butyrate, mmol/L	Iso-Butyrate, mmol/L	Valerate, mmol/L	Iso-Valerate, mmol/L	TVFA, mmol/L	A/P
Treatment								
CON	22.4 ^b^	8.58 ^b^	2.71 ^b^	0.31 ^b^	0.29	0.33 ^b^	34.6 ^b^	2.71 ^a^
P0.1	23.2 ^ab^	9.06 ^a^	2.80 ^ab^	0.32 ^ab^	0.30	0.35 ^b^	35.8 ^ab^	2.61 ^a^
P0.2	23.3 ^ab^	9.29 ^a^	2.84 ^a^	0.32 ^ab^	0.30	0.35 ^b^	36.2 ^a^	2.54 ^ab^
P0.4	24.1 ^a^	9.27 ^a^	2.92 ^a^	0.33 ^a^	0.30	0.37 ^a^	37.3 ^a^	2.64 ^a^
P0.55	23.8 ^a^	9.41 ^a^	2.89 ^a^	0.33 ^a^	0.31	0.36 ^ab^	37.1 ^a^	2.59 ^a^
Incubation time, h								
3	18.0 ^D^	5.79 ^D^	1.05 ^D^	0.16 ^C^	0.14 ^E^	0.12 ^D^	25.3 ^D^	3.15 ^A^
6	18.4 ^D^	7.52 ^C^	2.22 ^C^	0.16 ^C^	0.17 ^D^	0.13 ^D^	28.6 ^C^	2.45 ^C^
9	28.2 ^A^	10.3 ^B^	3.33 ^B^	0.33 ^B^	0.33 ^C^	0.31 ^C^	42.8 ^A^	2.75 ^B^
12	25.6 ^C^	10.1 ^B^	3.44 ^B^	0.47 ^A^	0.45 ^A^	0.58 ^B^	40.7 ^B^	2.54 ^C^
24	26.6^B^	11.9 ^A^	4.13 ^A^	0.48 ^A^	0.43 ^B^	0.62 ^A^	43.8 ^A^	2.20 ^D^
SEM	0.422	0.146	0.045	0.005	0.005	0.008	0.518	0.045
*p*-value								
TRT	0.049	0.001	0.013	0.004	0.159	0.027	0.018	0.103
linear	0.007	<0.001	0.001	<0.001	0.027	0.004	0.006	0.283
quadratic	0.181	0.047	0.122	0.642	0.677	0.312	0.091	0.120
Hour	<0.001	<0.001	<0.001	<0.001	<0.001	<0.001	<0.001	<0.001
TRT × Hour	0.930	0.975	0.673	0.866	0.817	0.555	0.740	0.440

^a–d^ Without a common superscript letter indicates significant (*p* < 0.05) changes among treatments. ^A–E^ Without a common superscript letter indicates significant (*p* < 0.05) changes among hours.

**Table 5 animals-13-00221-t005:** The effect of NFP on in vivo variables related to ruminal fermentation in cashmere goats.

Item	−1 h		6 h		SEM	*p*-Value		
	CON	NFP	CON	NFP		TRT	Hour	TRT × Hour
pH	6.93	6.86	6.65	6.69	0.055	0.882	<0.001	0.175
NH_3_-N, mg/dL	22.9	18.8	17.8	13.9	0.905	0.017	0.003	0.929
MCP, mg/dL	32.1	44.5	28.3	37.0	2.405	0.009	0.044	0.477
Protozoa, 10^4^/mL	34.1	29.2	53.6	36.8	2.610	0.010	0.002	0.126
Volatile fatty acids, mmol/L								
Acetate	21.5	25.6	26.4	32.1	1.129	0.016	0.001	0.503
Propionate	4.13	5.01	5.44	6.68	0.184	0.003	<0.001	0.392
Butyrate	3.64	5.12	5.78	6.76	0.375	0.068	<0.001	0.354
Iso-butyrate	0.517	0.590	0.427	0.467	0.012	0.010	<0.001	0.234
Valerate	0.287	0.346	0.349	0.406	0.019	0.075	0.035	0.968
Iso-valerate	0.674	0.758	0.427	0.484	0.018	0.028	<0.001	0.480
TVFA	30.7	37.5	38.7	46.9	1.436	0.006	<0.001	0.652
Acetate to propionate ratio	5.24	5.09	4.77	4.81	0.200	0.875	0.022	0.506

**Table 6 animals-13-00221-t006:** The effect of NFP on growth performance and apparent nutrient digestibility in cashmere goats.

Item	Treatment		SEM	*p*-Value
	CON	NFP		
Growth performance				
DMI, kg/d	1.35	1.63	0.071	0.033
BW beginning of the trial, kg	46.5	46.8	1.291	0.893
BW end of the trial, kg	47.5	48.6	1.277	0.579
ADG, kg	0.054 ^b^	0.097 ^a^	0.011	0.020
DMI:ADG	25.0 ^a^	16.8 ^ab^	2.770	0.065
Apparent nutrient digestibility, %				
Dry matter	77.31 ^a^	78.73 ^b^	0.301	0.008
Crude protein	81.03 ^a^	83.84 ^b^	0.692	0.015
Ether extract	84.99	84.11	0.579	0.308
Neutral detergent fiber	52.15	54.12	1.103	0.236
Acid detergent fiber	43.12	45.11	0.949	0.176
Calcium	38.60 ^a^	42.93 ^ab^	1.320	0.061
Phosphorus	43.79	48.25	4.922	0.540

^a,b^ Without a common superscript letter indicates significant (*p* < 0.05) changes between treatments.

**Table 7 animals-13-00221-t007:** Number of clean reads, operational taxonomic units (OTU) and alpha diversity indices of ruminal bacteria in Exp. 2.

Item	Treatment		SEM	*p*-Value
	CON	NFP		
Clean reads	44,208	43,091	2565	0.764
Sobs	800	774	23.2	0.436
Shannon	4.65	4.65	0.161	0.998
Simpson	0.047	0.039	0.016	0.725
Ace	971	980	21.9	0.778
Chao	981	981	24.2	0.989
Good’s coverage	0.9903 ^a^	0.9895 ^ab^	0.0003	0.064

^a,b^ Without a common superscript letter indicates significant (*p* < 0.05) changes between treatments.

**Table 8 animals-13-00221-t008:** Relative abundance of main bacterial phyla and genera in Exp. 2.

Item	CON	NFP	SEM	*p*-Value
phyla (proportion of total observations)				
Firmicutes	0.336 ^b^	0.647 ^a^	0.0411	<0.001
Bacteroidetes	0.526 ^a^	0.223 ^b^	0.0507	0.002
Saccharibacteria	0.036	0.050	0.0105	0.433
Actinobacteria	0.030 ^a^	0.003 ^b^	0.0056	0.044
Proteobacteria	0.012	0.021	0.0044	0.194
Synergistetes	0.012	0.019	0.0068	0.485
Tenericutes	0.018	0.004	0.0049	0.163
Verrucomicrobia	0.006	0.013	0.0038	0.263
family (proportion of total observations)				
Ruminococcaceae	0.186 ^b^	0.539 ^a^	0.0504	<0.001
Prevotellaceae	0.146	0.108	0.0322	0.420
Rikenellaceae	0.180 ^a^	0.048 ^b^	0.0227	0.005
Bacteroidales_BS11_gut_group	0.169 ^a^	0.039 ^ab^	0.0327	0.060
Unknown_Family_o__Unknown_Order_ c__Unknown_Class_p__Saccharibacteria	0.036	0.050	0.0045	0.433
Lachnospiraceae	0.048 ^a^	0.013 ^b^	0.0044	<0.001
Christensenellaceae	0.037	0.022	0.0066	0.166
Erysipelotrichaceae	0.015	0.032	0.0069	0.139
Bacteroidales_RF16_group	0.015	0.017	0.0031	0.801
Synergistaceae	0.012	0.019	0.0068	0.485
Veillonellaceae	0.015	0.006	0.0041	0.143
Bifidobacteriaceae	0.020	0.001	0.0037	0.124
Family_XIII	0.016 ^a^	0.003 ^b^	0.0026	0.028
Clostridiales_vadinBB60_group	0.006	0.013	0.0026	0.151
genera (proportion of total observations)				
Ruminococcus_1	0.027 ^b^	0.260 ^a^	0.0319	0.002
norank_f__Bacteroidales_BS11_gut_group	0.169 ^a^	0.039 ^ab^	0.0327	0.060
Rikenellaceae_RC9_gut_group	0.160 ^a^	0.046 ^b^	0.0224	0.011
Prevotella_1	0.114	0.091	0.0291	0.598
Candidatus_Saccharimonas	0.036	0.050	0.0105	0.433
Ruminococcaceae_UCG-002	0.022 ^b^	0.060 ^a^	0.0074	0.005
Ruminococcus_2	0.024	0.041	0.0126	0.369
Christensenellaceae_R-7_group	0.037	0.022	0.0066	0.172
Ruminococcaceae_NK4A214_group	0.033 ^a^	0.024 ^b^	0.0028	0.043
[Eubacterium]_coprostanoligenes_group	0.010	0.026	0.0056	0.150
norank_f__Erysipelotrichaceae	0.005 ^ab^	0.029 ^a^	0.0061	0.059
norank_f__Ruminococcaceae	0.009 ^b^	0.024 ^a^	0.0043	0.028
Prevotellaceae_UCG-003	0.024 ^a^	0.008 ^b^	0.0031	0.018
Bacteroidales_RF16_group	0.015	0.017	0.0048	0.801
Fretibacterium	0.012	0.019	0.0068	0.503
Ruminococcaceae_UCG-010	0.011	0.016	0.0025	0.149

^a,b^ Without a common superscript letter indicates significant (*p* < 0.05) changes between treatments. Abbreviations: CON, control treatment; NFP, treatment with NFP supplementation with 4 g/kg DM (0.40% NFP); SEM, standard error of mean.

## Data Availability

The data presented in this study are available on request from the corresponding author.

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
