# Peer review of "Dietary Polysaccharide-Rich Extract from Noni (Morinda citrifolia L.) Fruit Modified Ruminal Fermentation, Ruminal Bacterial Community and Nutrient Digestion in Cashmere Goats"

_animals, 2023, doi:10.3390/ani13020221_

Round 1

Reviewer 1 Report

Materials and Methods

131 Based on what did you add 0.1, 0.2, 0.4, 0.55%? Forage? Concentrate?

Table 2 DM's full name is not indicated. Also, the value is required if the DM is dry matter.

145 The body weight and standard deviation of the live animal should be provided.

157 in vitro italic

Result

318 In figure 2 of supplementary, there was no significant difference between treatment groups at 0 hour pH. It's good to mention this in manuscript.

321 You should indicate whether the factor is time or added level.

Discussion

This paper is expected to be meaningful in ruminant animal studies, where interest in the addition of plant extracts is increasing. In addition, research on ruminant microbes and protein utilization efficiency is being conducted in various aspects, so it is thought that the research results of this paper will be of great help to research on ruminants.

462 References are required

Supplementary

Table 1: famly->family

The standard error must have one more decimal place than the values (of CON and NFP). Decimal point in the data need to be corrected.

Information on CON and NFP should be written at the bottom (presence of addition, level of addition, etc.)

You should also list abbreviations such as SEM

Figure 2 nh3-n full name must also be indicated.

Reviewer 2 Report

The authors demonstrate that supplementation of feed with a complex, polysaccharide-rich fruit extract from Noni (Molinda citrifolia) substantially improved the ruminal fermentation in goats both in vitro and in vivo, and that part of the in vivo effect was due to increased feed intake. Additionally, the authors showed major differences in ruminal bacterial community composition in between noni extract-fed animals vs control (noni not fed) animals, along with lower levels of protozoa in the noni-fed group. Some of the differences in community composition were dramatic, particularly the dominance by genus Ruminococcus_1 in noni-fed group. Overall, this is an interesting and well-done study. The methods used were were comprehensively described. The manuscript is generally well-written. However, the reviewer has a few concerns that the authors need to address.

            The first concern regards the authors’ conclusion that 0.4% is the optimum level of noni extract inclusion in the ration, based on the in vitro experiment. It appears, though, that there was very little of a quantitative dose-response effect (see Comment to L331 below), and that 0.1% NFP yielded similar effects. The authors need to clarify this disparity between the text and the data in Table 3.    

            The second concern regards the use of stomach tubing to collect samples, and the use of the kit-based DNA isolation method. Stomach tubing collects primarily the liquid phase of ruminal contents; the authors are well aware of this, and in fact mention it the Discussion (L528-529). The reviewer suggests that the authors mention this more explicitly in the Methods section, as a caveat to the reader. Regarding the DNA isolation method, it is well-known that kit-based methods give low yields of microbial community DNA in the rumen of both cows and sheep (Henderson et al., doi:10.1371/journal.pone.0074787 ), and the isolated DNA is less representative of the ruminal community than that from high-yielding methods. This fact does not invalidate the present work, especially because the same method was used across samples. But the authors should mention the problem of kit-based isolation methods explicitly, again as a caution to the reader.

            The reviewer would like to make two additional points. First, the tremendously higher relative abundance of genus Ruminococcus_1 in the noni-supplemented goats is, in the reviewer’s opinion, one of the most exciting observations in the study. The observed relative abundance (26%) was ten-fold higher than in the control (as indicated in Table S1, which, by the way, is important enough to be included as a Table in the text, rather than as a Supplementary Table). As far as the reviewer knows, there are few dietary treatments that propel a genus of ordinarily modest abundance, to the degree of dominance in vivo that was observed here. The authors should make a bigger deal of this observation than they do.  Additionally, the NP substrate may be an ideal candidate for use in enrichment of Ruminococcus_1 in culture, which in turn should allow isolation of members of this genus in pure culture for more detailed physiological studies (which are largely lacking for this genus).

            The second point is that the authors’ comprehensive analysis of their polysaccharide-rich extract (L118-122 and Table 1) represents an impressive analytical feat that is almost unique in ruminant nutrition studies (where extracts or natural supplements receive little or no chemical characterization). The authors should expand on this by providing a supplementary table with a complete list of compounds that comprise each of the listed chemical classes, and their individual amounts. Many readers may like to know, for example, which specific polysaccharides are present (or even the monosaccharide composition of the polysaccharides), or what other natural products of known function may be present. Supplying this detailed information would really set this manuscript apart from other supplement feeding studies.

Specific comments:

L131: Clarification is needed here. Were the values on a w/v basis (e.g., 0.1% = 0.1 g of lyophilized extract/100 ml media), or on percent of dry matter basis (e.g., 0.1% = 0.1 g of lyophilized extract/100 g of medium solids)?

L202: If there were 7 groups of 2 goats, and within each group 1 goat received the NFP treatment (L199-202), should there not have been 7 (rather than 6) goats for each treatment?

L210: How were goats weighed “continuously” before feeding?

L281-285: Were singletons removed from the analysis?

L331: This statement is at odds with Table 3, which reports that all NFP treatments yielded similar values for all variables, except GP for P0.2.

L411-412: Here and elsewhere, when comparing across treatments in different animals, do not use the terms “increased” or “decreased”, as these terms imply changes within an individual animal over time. Instead, use “were greater than” or “were less than”. (Generally, the authors did properly use the latter terms, but not always).

L450-451: The authors should provide a literature citation to support this statement.

L485-489: See comment to L331 above. If, in fact, the 0.1% concentration was just as effective as the 0.4% concentration (as it was in vitro, and needs to be tested in a future feeding trial), the conclusion that the 0.4% level was optimal is premature.

L511-530: This is a very cogent explanation of the retention of fiber digestion upon reducing protozoal numbers, although the general phenomenon is typically called niche replacement.

Minor edits:

L114-115: Please rewrite. The polysaccharides were precipitated, not the solution.

L146: Change “Goats” to “Goat”.

L161: Change “addition” to “additions”.

L222: Change “was” to “were”.

L322: Change “went following” to “then”.

L324: Change “In” to “For”.

Table 6: The digestibilities are listed as “%”, but they are clearly fractions (e.g., 77% rather than 0.77%). Also, even though P values are stated for each variable, it would be useful to insert superscripted letters (a,b) next to treatment means that differed at P<0.05, to make the differences more visually apparent.

Fig.3: Suggest moving the labels for points C7 and N7 slightly to separate them from those of C3 and N4, respectively.

L399,404, 415: Change “were” to “are’ (present tense when referring to Figures or Tables).

L469: Change “battles” to “bottles”.

L471: Delete “While”.

Table S1: See comments to Table 6 above.

Reviewer 3 Report

The Manuscript ID animals-2014797 entitled Dietary polysaccharide-rich extract from noni (Morinda citrifolia L.) fruit, modified ruminal fermentation, ruminal bacterial community, and nutrient digestion in cashmere goats is an attempt to investigate the effect of polysaccharide-rich noni fruit extract (NFP) on ruminal fermentation, ruminal microbiota and nutrient digestion in cashmere goats combining in vitro and in vivo experiment. The submitted paper is interesting, especially in dairy goat fields study and tested alternative feed additives to prevent negative health aspects associated with the seasonal nutrition imbalance in a yard-feeding system due to limited access to green pasture. The submitted original paper is well written, and a lot of scientific work was done. Although the reviewer agrees with searching for new solutions (e.g. feed additives) to prevent the negative effect of seasonal nutrient imbalance, in the current form, the submitted manuscript should be improved and I recommend major revision. The details suggestions a submitted in the pdf file.

Round 2

Reviewer 2 Report

The reviewer thanks the authors for their revisions, and for their clear explanation in their Response to Reviewer letter. The reviewer believes the work would be an excellent contribution to the literature.  Two minor items for the authors' attention.

1) Remove "And" at the beginning of the sentence in L572.

2) Remove Supplementary Table S1, as it now appears as Table 8 in the main text.

Author Response

Dear reviewer,

Thank you very much for your thoughtful suggestions, which have greatly improved this article. We apologize for the omission during the last revision. We checked the review comments carefully and revised our copy according to them. A detailed response to your comments is attached below.

We would like to thank you for allowing us to resubmit a revised manuscript and we appreciate your time and consideration.

1) Remove "And" at the beginning of the sentence in L572.

Thank you for your careful advice. We have removed the "and".

2) Remove Supplementary Table S1, as it now appears as Table 8 in the main text.

Thank you for your advice. We are very sorry that we may have forgotten to upload the revised supplementary materials in the last round of replies. This table has been removed. See supplementary information for details.

Thank you again for your valuable comments.

Reviewer 3 Report

All suggestions sent by the reviewer were included in the manuscript. In the reviewer's opinion, the manuscript may be accepted for publishing in its current form.

Author Response

Dear Reviewer,

Thank you very much for your thoughtful suggestions that have helped improve this paper substantially. We would love to thank you for accepting the manuscript for publication in its current form. Moreover, we highly appreciate your time and consideration.

Kind regards,

Authors

  1. All suggestions sent by the reviewer were included in the manuscript. In the reviewer's opinion, the manuscript may be accepted for publishing in its current form.

au: Thanks for reviewer's recognition and positive comments on this study.

Again, many thanks for your valuable reviews.
